# CONTINUAL ACTIVE LEARNING

## ABSTRACT

While active learning (AL) improves the labeling efficiency of machine learning (by allowing models to query the labels of data samples), a major problem is that compute efficiency is decreased since models are typically retrained from scratch at each query round. In this work, we develop a new framework that circumvents this problem by biasing further training towards the recently labeled sets, thereby complementing existing work on AL acceleration. We employ existing and novel replay-based Continual Learning (CL) algorithms that are effective at quickly learning new samples without forgetting previously learned information, especially when data comes from a shifting or evolving distribution. We call this compute-efficient active learning paradigm *"Continual Active Learning" (CAL)*. We demonstrate that standard AL with warm starting fails, both to accelerate training, and that naive fine-tuning suffers from catastrophic forgetting due to distribution shifts over query rounds. We then show CAL achieves significant speedups using a plethora of replay schemes that use model distillation, and that select diverse/uncertain points from the history, all while maintaining performance on par with standard AL. We conduct experiments across many data domains, including natural language, vision, medical imaging, and computational biology, each with very different neural architectures (Transformers/CNNs/MLPs). CAL consistently provides a 2–6x reduction in training time, thus showing its applicability across differing modalities.

## 1 INTRODUCTION

While neural networks have been immensely successful in a variety of different supervised settings, most deep learning approaches are data-hungry and require significant amounts of computational resources. From a large pool of unlabeled data, active learning (AL) approaches select subsets of points to label by imparting the learner with the ability to query a human annotator. Such methods incrementally add points to the pool of labelled samples by 1) training a model from scratch on the current labelled pool and 2) using some measure of model uncertainty and/or diversity to select a set of points to query the annotator (Settles, 2009; 2011; Wei et al., 2015; Ash et al., 2020; Killamsetty et al., 2021). AL has been shown to reduce the amount of data required for training, but can still be computationally expensive to employ since it requires retraining the model, typically from scratch, when new points are labelled at each round.

A *simple* way to tackle this problem is to warm start the model parameters between rounds to reduce the convergence time. However, the observed speedups tend to still be limited since the model must make several passes through an ever-increasing pool of data. Moreover, warm starting alone in some cases can hurt generalization, as discussed in Ash & Adams (2020) and Beck et al. (2021). Another extension to this is to solely train on the newly labeled batch of examples to avoid re-initialization. However, as we show in Section 3.3, naive fine-tuning fails to retain accuracy on previously seen examples since the distribution of the query pool may drastically change with each round.

This problem of *catastrophic forgetting* while incrementally learning from a series of new tasks with shifting distribution is a central question in another paradigm called Continual Learning (CL) (French, 1999; McCloskey & Cohen, 1989; McClelland et al., 1995; Kirkpatrick et al., 2017c). CL has recently gained popularity, and many algorithms have been introduced to allow models to quickly adapt to new tasks without forgetting (Riemer et al., 2018; Lopez-Paz & Ranzato, 2017; Chaudhry et al., 2019; Aljundi et al., 2019b; Chaudhry et al., 2020; Kirkpatrick et al., 2017b).

In this work, we propose Continual Active Learning (CAL), which applies continual learning strategies to accelerate batch active learning. In CAL, we propose applying CL to enable the model to learn the newly labeled points without forgetting previously labeled points while using past samples efficiently using *replay-based* methods. As such, we observe that CAL methods attain significant speedups over standard AL in terms of training time. Such speedups are beneficial for the following reasons:

- As neural networks grow in size (Shoeybi et al., 2019), the environmental and financial costs to train these models increase as well (Bender et al., 2021; Dhar, 2020; Schwartz et al., 2020). Reducing the number of gradient updates required for AL will help mitigate such costs, especially with large-scale models.

- Reducing the compute required for AL makes AL-based tools more accessible for deployment on edge computing platforms, IoT, and other low-resource devices (Senzaki & Hamelain, 2021).

- Developing new AL algorithms/acquisition functions, or searching for architectures as done with NAS/AutoML, that are well-suited *specifically* for AL can require hundreds or even thousands of runs. Since CAL's speedups are agnostic to the AL algorithm and the neural architecture, such experiments can be significantly sped up.

The importance of speeding up the training process in machine learning is well recognized and is evidenced by the plethora of optimized machine learning training literature seen in the computing systems community (Zhihao Jia & Aiken.; Zhang et al., 2017; Zheng et al., 2022).

In addition, CAL demonstrates a practical application for CL methods. Many of the settings used to benchmark CL methods in recent works are somewhat contrived and unrealistic. Most CL works consider the class/domain incremental setting, where only the samples that belong to a subset of the set of classes/domains of the original dataset are available to the model at any given time. This setting rarely occurs in practice, representing the worst-case scenario and therefore should not be the only benchmark upon which CL methods are evaluated. We posit that the validity of future CL algorithms may be determined based on their performance in the CAL setting in addition to their performance in existing benchmarks.

To the best of our knowledge, this application of CL algorithms for batch AL has never been explored. Our contributions can be summarized as follows: (1) We first demonstrate that active learning can be viewed as a continual learning problem and propose the CAL framework; (2) we benchmark several existing CL methods (CAL-ER, CAL-DER, CAL-MIR) as well as novel methods (CAL-SD, CAL-SDS2) and evaluate them on several datasets based on the accuracy/speedup they can attain over standard AL.

## 2 RELATED WORK

Active learning has demonstrated label efficiency (Wei et al., 2015; Killamsetty et al., 2021; Ash et al., 2020) over passive learning. In addition to these empirical advances there has been extensive work on theoretical aspects as well over the past decade (Hanneke, 2009; 2007; Balcan et al., 2010) where Hanneke (2012) shows sample complexity advantages over passive learning in noise-free classifier learning for VC classes. However, recently there has been an interest in speeding up active learning because most deep learning involves networks with a huge numbers of parameters.

Kirsch et al. (2019); Pinsler et al. (2019); Sener & Savarese (2018) aim to reduce the number of query iterations by having large query batch sizes. However, they do not exploit the learned models from previous rounds for the subsequent ones and are therefore complementary to CAL. Works such as Coleman et al. (2020a); Ertekin et al. (2007); Mayer & Timofte (2020); Zhu & Bento (2017) speed up the selection of the new query set by appropriately restricting the search space or by using generative methods. These works can be easily integrated into our framework because CAL works on the training side of active learning, not on the query selection. On the other hand, Lewis & Catlett (1994); Coleman et al. (2020b); Yoo & Kweon (2019) use a smaller proxy model to reduce computation overhead, however, they still follow the standard active learning protocol, and therefore can be accelerated when integrated with CAL.

Lastly, there exist a few prior works that explore continual/transfer learning and active learning in the same context. Perkonigg et al. (2021) propose an approach that allows active learning algorithms to be applied to data streams in the context of medical imaging, by introducing a module that detects domain shifts. This differs from our work which uses algorithms that prevent catastrophic forgetting, to accelerate active learning. Zhou et al. (2021) consider a setting in which standard active learning is used to finetune a pre-trained model, and uses transfer learning to do so. Thus, this work does not consider continual learning and active learning in the same setting and is therefore not related to our work.

On preventing catastrophic forgetting, in this work, we mostly focus on the replay-based algorithms that are currently state-of-the-art methods in continual learning. However, as demonstrated in Section 3.3 on how active learning rounds can be seen as continual learning, one can apply other methods such as EWC (Kirkpatrick et al., 2017a), structural regularization (Li et al., 2021) or functional regularization based methods as well. (Titsias et al., 2020).

# 3 METHODS

## 3.1 BATCH ACTIVE LEARNING

Define $[n] = \{1, ..., n\}$, and let $\mathcal{X}$ and $\mathcal{Y}$ denote the input and output domains respectively. AL typically starts with an unlabelled dataset $\mathcal{U} = \{x_i\}_{i \in [n]}$, where each $x_i \in \mathcal{X}$. The AL setting allows the model $f$, with parameters $\theta$, to query a user for labels for any $x \in \mathcal{U}$, but the total number of labels is limited to a budget $b$, where $b < n$. Throughout the work, we consider classification tasks so the output of $f(x; \theta)$ is a probability distribution over classes. The goal of AL is to ensure that $f$ can attain low error when trained only on the set of $b$ labelled points.

Algorithm 1 details the general AL procedure. Lines 3-6 construct the seed set $\mathcal{D}_1$, by randomly sampling a subset of points from $\mathcal{U}$ and labelling them. Lines 7-14 iteratively expand the labelled set for $T$ rounds by training the model from a random initialization on $\mathcal{D}_t$ until convergence and selecting $b_t$ points (where $\sum_{t \in [T]} b_t = b$) from $\mathcal{U}$ based on some selection criteria that is dependent on $\theta_t$. The selection criteria generally selects samples based model uncertainty and/or diversity (Lewis & Gale, 1994; Dagan & Engelson, 1995; Settles; Killamsetty et al., 2021; Wei et al., 2015; Ash et al., 2020; Sener & Savarese, 2017). In this work, we primarily consider uncertainty sampling Lewis & Gale (1994); Dagan & Engelson (1995); Settles, though we also test other selection criteria in Section A in the Appendix.

---

**Algorithm 1**

1: **procedure** ACTIVELEARNING($f, \mathcal{U}, b_{1:T}, T$)
2:      $t \leftarrow 1, \mathcal{L} \leftarrow \emptyset$                                              ▷ Initialize
3:      $\mathcal{U}_t \sim \mathcal{U}$                                   ▷ Draw $b_1$ samples from $\mathcal{U}$
4:      $\mathcal{D}_t \leftarrow \{(x_i, y_i) | x_i \in \mathcal{U}_t\}$                         ▷ Provide labels
5:      $\mathcal{U} \leftarrow \mathcal{U} \setminus \mathcal{U}_t$
6:      $\mathcal{L} \leftarrow \mathcal{L} \cup \mathcal{D}_t$
7:      **while** $t \leq T$ **do**
8:          Randomly initialize $\theta_{init}$
9:          $\theta_t \leftarrow \text{Train}(f, \theta_{init}, \mathcal{L})$
10:         $\mathcal{U}_t \leftarrow \text{Select}(f, \theta_t, \mathcal{U}, b_t)$           ▷ Select $b_t$ points from $\mathcal{U}$ based on $\theta_t$
11:         $\mathcal{D}_t \leftarrow \{(x_i, y_i) | x_i \in \mathcal{U}_t\}$
12:         $\mathcal{U} \leftarrow \mathcal{U} \setminus \mathcal{U}_t; \mathcal{L} \leftarrow \mathcal{L} \cup \mathcal{D}_t; t \leftarrow t + 1$
13:      return $\mathcal{L}$

---

**Uncertainty Sampling**    is a widely-used practical AL method that selects $\mathcal{U}_t = \{x_1, ..., x_{b_t}\}$ to label from $\mathcal{U}$ by choosing the samples that maximize a notion of model uncertainty. We consider entropy (Dagan & Engelson, 1995) as the uncertainty metric, so if $h(x) \triangleq - \sum_{i \in [k]} f(x; \theta)_i \log f(x; \theta)_i$, then $\mathcal{U}_t \in \arg\max_{\mathcal{A}: |\mathcal{A}| = b_t} \sum_{x \in \mathcal{A}} h(x)$.

## 3.2 CONTINUAL LEARNING

We define $\mathcal{D}_{1:n} = \bigcup_{i \in [n]} \mathcal{D}_i$. In CL, the dataset consists of $T$ tasks $\{\mathcal{D}_1, ..., \mathcal{D}_T\}$ that are presented to the model sequentially, where $\mathcal{D}_t = \{(x_i, y_i)\}_{i \in [n_t]}$ and $n_t$ is the cardinality of $\mathcal{D}_t$. At time $t \in [T]$, the data/label pairs are sampled from the current task $(x, y) \sim \mathcal{D}_t$, and the model generally has only limited access to the history $\mathcal{D}_{1:t-1}$. The CL objective is to efficiently adapt the model to $\mathcal{D}_t$ while ensuring that performance on previously learnt tasks $\mathcal{D}_{1:t-1}$ does not degrade appreciably. Ideally, given a loss function $\ell : \mathcal{X} \times \mathcal{Y} \mapsto \mathbb{R}$, initial parameters $\theta_{t-1}$, and a model $f$, $\theta_t$ can be obtained by solving the CL optimization problem (Aljundi et al., 2019b; Chaudhry et al., 2019; Lopez-Paz & Ranzato, 2017):

$$\arg\min_{\theta} \mathbb{E}_{(x,y)\sim\mathcal{D}_t} \ell(y, f(x; \theta))$$

$$\text{s.t} \quad \mathbb{E}_{(x',y')\sim\mathcal{D}_{1:t-1}} \ell(y', f(x'; \theta))) \leq \mathbb{E}_{(x',y')\sim\mathcal{D}_{1:t-1}} \ell(y', f(x'; \theta_{t-1})))$$

In this work, we focus on replay based CL techniques which attempt to approximately solve the CL optimization problem by using samples from $\mathcal{D}_{1:t-1}$ to regularize the model while adapting to $\mathcal{D}_t$.

Algorithm 2 outlines the general replay-based CL algorithm, in which the objective is to adapt $f$ parametrized by $\theta_0$ to $\mathcal{D}$ while using samples from the history $\mathcal{M}$. Inside the training loop, $\mathcal{B}_{\text{current}}$ consists of $m$ points randomly sampled from $\mathcal{D}$. $\mathcal{B}_{\text{replay}}$ consists of $m'$ points that are chosen based on some customizable selection criteria from $\mathcal{M}$. In line 6, $\theta_t$ is computed based on some update rule that utilizes both $\mathcal{B}_{\text{replay}}$ and $\mathcal{B}_{\text{current}}$. Note that many CL works also consider the problem of selecting which samples should be retained in $\mathcal{M}$, which is relevant in the scenario where $\mathcal{D}_{1:T}$ is too large to store in memory or when $T$ is unknown (Aljundi et al., 2019b). However, this constraint does not apply to the CAL setting, so in the subsequent sections we consider $\mathcal{M} = \mathcal{D}_{1:t-1}$.

---

**Algorithm 2**

---

1: **procedure** CONTINUALTRAIN($f, \theta_0, \mathcal{D}, \mathcal{M}, m, m'$)
2:      $t \leftarrow 1$
3:      **while** not converged **do**
4:          $\mathcal{B}_{\text{current}} \leftarrow \{(x_i, y_i)\}_{i=1}^m \sim \mathcal{D}$            ▷ Sample $m$ points from current task
5:          $\mathcal{B}_{\text{replay}} \leftarrow \text{Select}(f, \theta_{t-1}, \mathcal{M}, m')$         ▷ Sample replay $m'$ points from history
6:          $\theta_t \leftarrow \text{Update}(f, \theta_{t-1}, \mathcal{B}_{\text{current}}, \mathcal{B}_{\text{replay}})$
7:          $t \leftarrow t + 1$
8:      return $\theta_t$

---

## 3.3 ACTIVE LEARNING AS CONTINUAL LEARNING

A clear inefficiency of standard AL stems from the fact that the model $f$ must be retrained from scratch on the labelled pool at every round. In this work, we employ CL-inspired techniques to adapt to the newly labelled points, while significantly reducing the number of updates needed on samples labelled in previous rounds.

We demonstrate that catastrophic forgetting indeed occurs in AL, when a model is fine-tuned only on the newly labelled points at every round. In Figure 1, task $t$ indicates the set of points from the training dataset that were selected at the round $t$ of querying based on entropy sampling. On the y-axis, we report the accuracy of each set immediately after the model has been fine-tuned on the points that were just labelled at a particular round.

It is evident that the model forgets old information from the precipitous drops in performance for task $t-1$ as soon as the model is adapted to new task $t$ when points are added to the labelled set. Note that task 1, after the initial drop, tends to increase in performance in the subsequent AL rounds since the points belonging to the initial round are chosen uniformly at random (shown in Algorithm 1) and thus is an unbiased estimate of the full dataset. This trend is generally not present in any of the later tasks, which are sampled from distributions that are conditioned on the model parameters $\theta_t$. It is also interesting to note that the model performs considerably worse on all of the tasks (aside from task 1) than it does on the test set, despite the fact that the model has been trained on the labelled

pool. This experiment suggests that 1) the distribution of each task $t > 1$ is distinct from the true data distribution and 2) techniques designed to combat catastrophic forgetting are necessary in order to effectively incorporate new information between successive AL rounds.

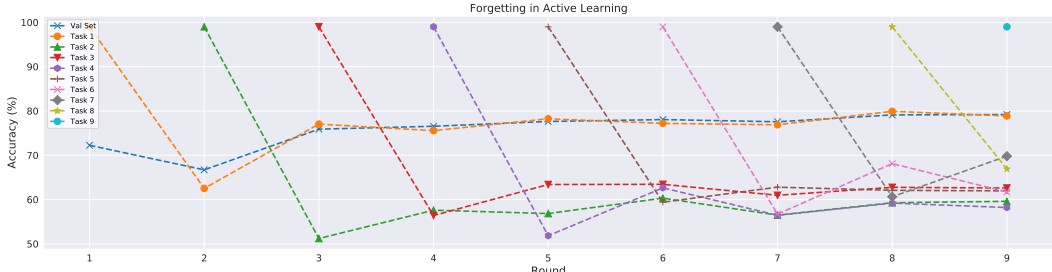

Figure 1: This figure shows the performance of a ResNet-18 on CIFAR-10, in the active learning setting where the model is only trained on newly labelled points. At each round, 5% of the full dataset is added to the labelled pool.

---

**Algorithm 3**

1: **procedure** CAL($f, \mathcal{U}, b, T, m, m'$)
2:      $t \leftarrow 1, \mathcal{L} \leftarrow \emptyset$                                        $\triangleright$ Initialize
3:      $\mathcal{U}_t \sim \mathcal{U}$                               $\triangleright$ Draw $b_1$ samples from $\mathcal{U}$
4:      $\mathcal{D}_t \leftarrow \{(x_i, y_i) | x_i \in \mathcal{U}_t\}$                        $\triangleright$ Provide labels
5:      $\mathcal{U} \leftarrow \mathcal{U} \setminus \mathcal{U}_t$
6:      $\mathcal{L} \leftarrow \mathcal{L} \cup \mathcal{D}_t$
7:      **while** $t \leq T$ **do**
8:          $\theta_t \leftarrow$ ContinualTrain($f, \theta_{t-1}, \mathcal{D}_t, \mathcal{D}_{1:t-1}, m, m'$)
9:          $\mathcal{U}_{t+1} \leftarrow$ Select($f, \theta_t, \mathcal{U}, b_t$)          $\triangleright$ Select $b_t$ points from $\mathcal{U}$ based on $\theta_t$
10:          $\mathcal{D}_{t+1} \leftarrow \{(x_i, y_i) | x_i \in \mathcal{U}_{t+1}\}$
11:          $\mathcal{U} \leftarrow \mathcal{U} \setminus \mathcal{U}_{t+1}; \mathcal{L} \leftarrow \mathcal{L} \cup \mathcal{D}_{t+1}; t \leftarrow t + 1$
12:      return $\mathcal{L}$

---

To ameliorate the problem of catastrophic forgetting, we use CL techniques. The continual active learning (CAL) approach is shown in Algorithm 3. The key difference of CAL from standard AL (Algorithm 1) can be found in line 8. Instead of standard training, replay-based CL is used to adapt $f$ to $\mathcal{D}_t$ while retaining performance on $\mathcal{D}_{1:t-1}$. The speedup comes from two points: 1) the number of gradient updates computed for samples from $\mathcal{D}_{1:t-1}$ is less than that of samples in $\mathcal{D}_t$ for reasonable choices of $m'$ and 2) the model tends to converge faster since its parameters are warm-started. We compare several CAL methods and assess their performance based on their performance on the test set and the speedup they attain compared to standard AL. In the rest of the section

$$\mathcal{L}_c \triangleq \mathbb{E}_{(x,y) \sim \mathcal{B}_{\text{current}}} [\ell(y, f(x; \theta))] \tag{1}$$

**Experience Replay (CAL-ER)** is the simplest and oldest replay-based method (Ratcliff, 1990; Robins, 1995). In this approach, $\mathcal{B}_{\text{current}}$ and $\mathcal{B}_{\text{replay}}$ are interleaved to create a minibatch $\mathcal{B}$ of size $m + m'$ and $\mathcal{B}_{\text{replay}}$ is chosen uniformly at random from $\mathcal{D}_{1:t-1}$. The parameters $\theta$ of model $f$ are updated based on the gradient computed on $\mathcal{B}$.

**Maximally Interferred Retrieval (CAL-MIR)** addresses the problem of selecting samples from $\mathcal{D}_{1:t-1}$, by choosing the $m'$ points that are most likely to be forgotten (Aljundi et al., 2019a). Given a batch of $m$ labelled samples $\mathcal{B}_{\text{current}}$ sampled from $\mathcal{D}_t$ and model parameters $\theta$, $\theta_v$ is computed by taking a "virtual" gradient step i.e. $\theta_v = \theta - \eta \nabla \mathcal{L}_c$ where $\eta$ is the learning rate. Then for every example $x$ in the history, $s_{MIR}(x) = \ell(f(x; \theta), y) - \ell(f(x; \theta_v), y)$ or the change in loss after taking a single gradient step is computed. The $m'$ samples with the highest $s_{MIR}$ score are selected to form

$\mathcal{B}_{\text{replay}}$. $\mathcal{B}_{\text{current}}$ and $\mathcal{B}_{\text{replay}}$ are concatenated together to form the minibatch (as in CAL-ER), upon which the gradient update is computed. In practice, selection is done on a random subset of $\mathcal{D}_{1:t-1}$ for speed.

**Dark Experience Replay (CAL-DER)** uses a distillation based approach to regularize updates (Buzzega et al., 2020). Suppose $g(x;\theta)$ denotes the presoftmax logits of classifier $f(x;\theta)$ i.e $f(x;\theta) = \text{softmax}(g(x;\theta))$. In DER, every $x' \in \mathcal{D}_{1:t-1}$ has an associated $z'$ which corresponds to the logits produced by the model at the end of the task when $x$ was first observed. In other words, if $x' \in \mathcal{D}_{t'}$, then $z' \triangleq g(x';\theta_{t'}^*))$ where $t' \in [t-1]$ and $\theta_{t'}^*$ are the parameters obtained after round $t'$. DER minimizes $\mathcal{L}_{DER}$ as expressed below:

$$\mathcal{L}_{DER} \triangleq \mathcal{L}_c + \mathop{\mathbb{E}}_{(x',y',z')\sim\mathcal{B}_{\text{replay}}} \left[ \alpha \left\| g(x';\theta) - z' \right\|_2^2 + \beta\, \ell(y', f(x';\theta)) \right], \qquad (2)$$

where $\mathcal{B}_{\text{current}}$ is a batch sampled from $\mathcal{D}_t$, $\mathcal{B}_{\text{replay}}$ is a batch sampled from $\mathcal{D}_{1:t-1}$, and $\alpha$ and $\beta$ are tuneable hyperparameters. The first term ensures that samples from the current task are classfied correctly. The second term consists of a classification loss and a mean squared error (MSE) based distillation loss that are applied on samples from the history.

**Scaled Distillation (CAL-SD)** is a new CL approach we propose in this work specifically tailored towards the CAL setting. SD addresses the stability-plasticity dilemma that is commonly found in both biological and artificial neural networks (Abraham & Robins, 2005; Mermillod et al., 2013). A network is *stable* if it can effectively retain past information but cannot adapt to new tasks efficiently, whereas a network that is *plastic* can quickly learn new tasks but is prone to forgetting. The trade-off between stability and plasticity is a well-known constraint in CL (Mermillod et al., 2013). In the context of CAL, we would like the model to be plastic during the early rounds and stable during the later rounds. We apply this intuition to develop SD, which minimizes $\mathcal{L}_{SD}$ at round $t$ as shown below:

$$\mathcal{L}_{\text{replay}} \triangleq \mathop{\mathbb{E}}_{(x',y',z')\sim\mathcal{B}_{\text{replay}}} \left[ \alpha\, D_{\text{KL}}\left(\text{softmax}(z') \,\|\, f(x';\theta)\right) + (1-\alpha)\,\ell\left(y', f(x';\theta)\right) \right], \qquad (3)$$

$$\mathcal{L}_{SD} \triangleq \lambda_t\, \mathcal{L}_c + (1-\lambda_t)\,\mathcal{L}_{\text{replay}}, \qquad (4)$$

where,

$$\lambda_t \triangleq \frac{1}{1 + \frac{|\mathcal{D}_{1:t-1}|}{|\mathcal{D}_t|}} \qquad (5)$$

Similar to CAL-DER, $\mathcal{L}_{\text{replay}}$ is a sum of two losses: a distillation loss and a classification loss. The distillation loss in $\mathcal{L}_{\text{replay}}$ minimizes the KL divergence between the posterior probabilities produced by $f$ and $\text{softmax}(z')$, where $z'$ is defined in the DER section. We use a KL divergence term instead of a MSE loss on the logits, so that the distillation loss and the classification losses are on the same scale. $\alpha \in [0,1]$ is a tuneable hyperparameter.

$\mathcal{L}_{SD}$ is a convex combination of the classification loss on the current task and $\mathcal{L}_{\text{replay}}$. The weight of each term is determined adaptively by the stability/plasticity trade-off term $\lambda_t$. Higher values of $\lambda_t$ indicate higher model plasticity, since minimizing the classification error of samples from the current task is prioritized. $\mathcal{D}_{1:t-1}$ increases with $t$, $\lambda_t$ decreases and the model becomes more stable in the later rounds of training.

**Scaled Distillation w/ Submodular Sampling (CAL-SDS2)** CAL-SDS2 is another a new CL approach we introduce in this work. CAL-SDS2 uses CAL-SD to regularize the model and utilizes submodular sampling to select a diverse set of points from the history to replay. Submodular functions are well suited to capture notions of diversity and representativeness (Lin & Bilmes, 2011; Wei et al., 2015; Bilmes, 2022), and the greedy algorithm can approximately maximize a monotone submodular function up to a $1 - e^{-1}$ factor guarantee (Fisher et al., 1978; Minoux, 1978; Mirzasoleiman et al., 2015). We define a submodular function $G$ below:

$$G(\mathcal{S}) \triangleq \sum_{x_i \in \mathcal{A}} \max_{x_j \in \mathcal{S}} w_{ij} + \lambda \log\left(1 + \sum_{x_i \in \mathcal{S}} h(x_i)\right), \qquad (6)$$

The first term of $G$ is the facility location function, where $w_{ij}$ is a similarity score between samples $x_i$ and $x_j$. In our experiments, $w_{ij} = \exp\left(-\|z_i - z_j\|^2/2\sigma^2\right)$ where $z_i$ is the penultimate layer representation of model $f$ for $x_i$ and $\sigma$ is a hyperparameter. The second term is a a concave over modular function (Liu et al., 2013) and $h(x_i)$ is some measure of model uncertainty. In order to speed up SDS2, we randomly subsample from the history before performing submodular maximization so $\mathcal{S} \subset \mathcal{A} \subset \mathcal{D}_{1:t-1}$. The objective of CAL-SDS2 is to ensure that the set of samples that are replayed are both difficult and diverse, similar to the motivation of the heuristic employed in Wei et al. (2015).

## 4 RESULTS

In this section, we evaluate the validation performance of the model when we train on different fractions $(b/n)$ of the full dataset. We compute the factor speedup attained by a CAL method by dividing the runtime of AL over the runtime of the CAL method. We test the CAL methods on a variety of different datasets spanning multiple modalities. The two methods that do not utilize CAL are AL w/ WS (Active Learning with Warm Starting) and AL. We plot speedup vs mean test accuracy (computed over three random seeds) at different labelling budgets $(b/n)$ for each of the five datasets we consider in this work. Qualitatively, methods that are plotted towards the top right corners are preferable. The results are also available in tabular form in Appendix A. We adapt the AL framework proposed in Beck et al. (2021) for all experiments presented in this section. In the main paper, we show results for uncertainty sampling based acquisition function, but provide results on other acquisition functions as well in Appendix B. Our objective is to demonstrate 1) at least one CAL method exists that can match or outperform a standard active learning technique while achieving a significant speedup for every budget and dataset and 2) models that have been trained using a CAL method behave no differently than standard models.

### 4.1 EXPERIMENTAL SETUP

**FMNIST**   The FMNIST dataset is a dataset consisting of 70,000 $28\times28$ grayscale images of fashion items belonging to 10 classes (Xiao et al., 2017). A ResNet-18 architecture (He et al., 2016) and SGD is used. We apply data augmentations, as in Beck et al. (2021), consisting of random horizontal flips and random croppings. On this dataset, we find that a CAL method matches or outperforms the performance of standard AL in every setting we test 2.

**CIFAR-10**   CIFAR-10 consists of 60,000 $32\times32$ color images with 10 different categories (Krizhevsky, 2009). We use a ResNet-18 and use the SGD optimizer for all CIFAR-10 experiments. We apply data augmentations consisting of random horizontal flips and random croppings. From the results shown in Figure 3, there is at least one CAL method that outperforms standard AL for every budget that we examine.

**MedMNIST**   We use the DermaMNIST dataset within the MedMNIST database (Yang et al., 2021a;b) for performance evaluation of CAL on medical imaging modalities. DermaMNIST consists of 3-color channel dermatoscope images of 7 different skin diseases, originally obtained from Codella et al. (2019); Tschandl et al. (2018). A ResNet-18 architecture is used for all DermaMNIST experiemnts. All results are shown in Figure 4.

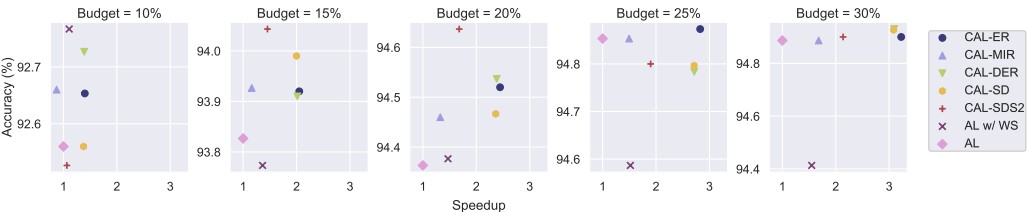

Figure 2: FMNIST Results

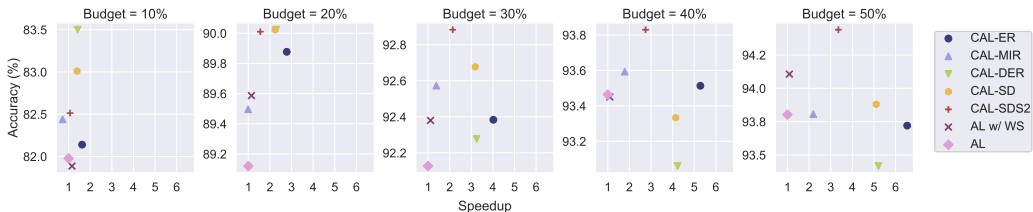

Figure 3: CIFAR-10 Results

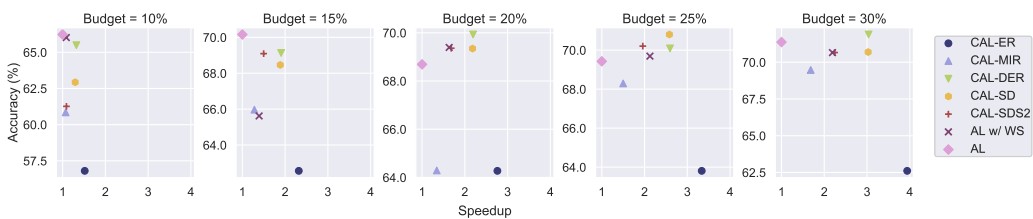

Figure 4: MedMNIST Results

**Amazon Polarity**  Similar to Coleman et al. (2020b), we use Amazon Polarity Review (Zhang et al., 2015) dataset, which is an NLP dataset consisting of reviews from Amazon and their corresponding star-ratings (5 classes). We consider total unlabelled pool of size 2M sentences and use VDCNN-9 Schwenk et al. (2017) architecture, trained with Adam optimizer. As observed from Figure 5, CAL methods achieve speedups while having competitive performance with standard AL procedure.

**COLA**  (Warstadt et al., 2018) is an another commonly used NLP dataset, which was recently considered in Active Learning setting (Ein-Dor et al., 2020). It aims to check linguistic acceptability of a sentence, that is, binary classification. We use BERT (Devlin et al., 2019) backbone trained with Adam optimizer. We consider an unlabled pool of size 7000 and remaining as test; similar to Ein-Dor et al. (2020) we use entropy sampling for the acquisition function and report accuracy. Figure 6 reports the performance and speedup of CAL methods with increasing budget, which shows their competitive performance with standard AL procedure.

**Single-Cell Cell Type Identity Classification**  Recent single-cell RNA sequencing (scRNA-seq) technologies has enabled large-scale characterization of hundreds of thousands to millions of cells in complex tissues, and accurate cell type annotation is a crucial step in the study of such datasets. To this end, several deep learning models have been proposed to automatically label new scRNA-seq datasets  (Xie et al., 2021). The HCL dataset is a highly class-imbalanced dataset that consists of scRNA-seq data for 562,977 cells across 63 cell types represented in 56 human tissues.  (Han et al., 2020). The data is divided into training, validation and test sets via an 80/10/10 split whilst ensuring similar class proportions across splits. We use the ACTINN model  (Ma & Pellegrini, 2019), a four-layer multi-layer perceptron that predicts the cell-type for each cell given its expression of 28832 genes, and use the SGD optimizer for all experiments. From the results shown in Figure 7, the majority of the CAL methods outperforms standard AL for every subset size that we examine.

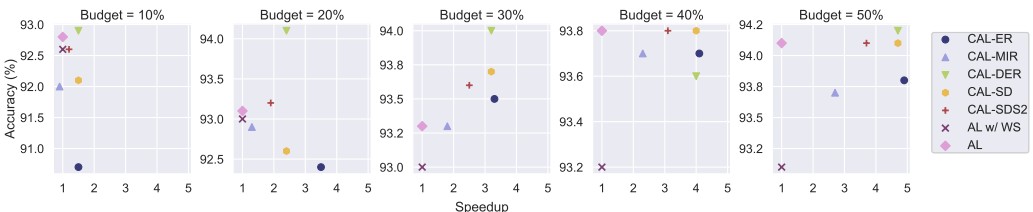

Figure 5: Amazon Polarity Results

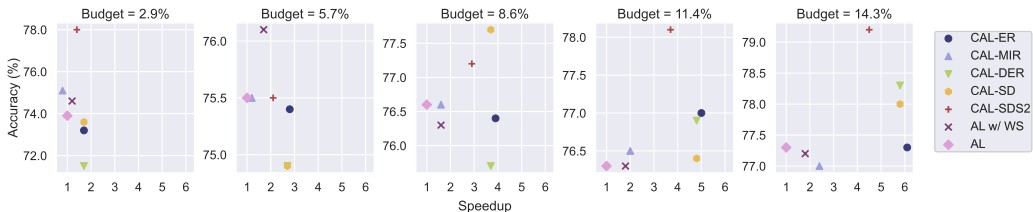

Figure 6: COLA Results

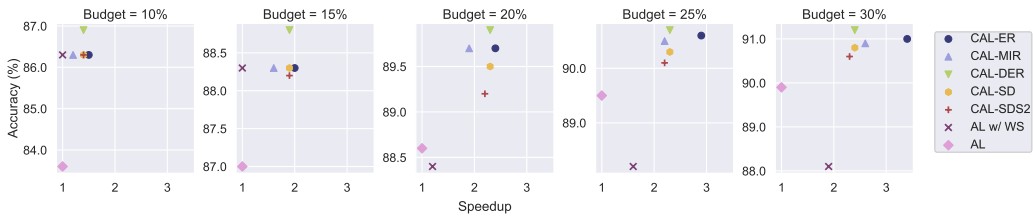

Figure 7: Single-Cell Cell-Type Identity Classification Results

## 4.2 SCORE CORRELATION BETWEEN STANDARD AND CAL MODELS

We test whether or not CAL models behave the same way as models that have been trained using standard AL. Specifically, we assess the degree to which the uncertainty scores of CAL models are correlated with standard models. In Figure 8, we show the pairwise correlation between all the entropy scores of the models we used in the FMNIST and CIFAR-10 experiments at the end of training (after training on 50% of the data). From the results, it is evident that the all the entropy scores are positively correlated, providing an explanation as to why CAL models are able to perform on par with standard models.

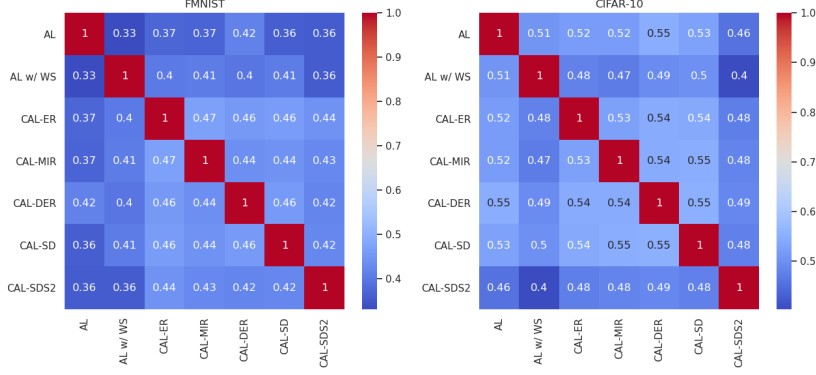

Figure 8: The correlation of entropy scores on the test set between models trained using AL/CAL at the end of FMNIST and CIFAR-10 experiments is shown.

## 5 CONCLUSION

In this work, we proposed the framework of CAL and demonstrated its efficacy in speeding up AL across multiple datasets by applying techniques adapted from CL. Across vision, natural language, medical imaging, and biological datasets, we observe that there is always a CAL method that either matches or outperforms standard AL while achieving considerable speedups. Since CAL is independent of model architecture and AL strategy, this framework is applicable to a broad range of settings. Furthermore, CAL provides a novel application for CL so future CL algorithms can be assessed based on their performance on CAL as well as other existing CL benchmarks.

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

## A  ADDITIONAL EXPERIMENTAL DETAILS ON MAIN RESULTS

### A.1  RESULTS IN TABULAR FORM

In this section, we report all results presented in Section 3.1 and Section 3.2 in tabular form. All methods highlighted in blue are methods that use CAL.

| Method | Test Accuracy (%) | | | | | Factor Speedup | | | | |
|---|---|---|---|---|---|---|---|---|---|---|
| | 10% | 15% | 20% | 25% | 30% | 10% | 15% | 20% | 25% | 30% |
| CAL-ER | 92.6 ±0.1 | 93.9 ±0.2 | 94.5 ±0.1 | **94.9** ±0.2 | **94.9** ±0.2 | 1.5× | 1.4× | 2.0× | 2.4× | 2.8 × |
| CAL-MIR | 92.6 ±0.3 | 93.9 ±0.2 | 94.5 ±0.0 | **94.9** ±0.1 | **94.9** ±0.0 | 0.9 × | 1.2× | 1.3× | 1.5× | 1.7× |
| CAL-DER | **92.7** ±0.1 | 93.9 ±0.1 | 94.5 ±0.1 | 94.8 ±0.2 | **94.9** ±0.1 | 1.4 × | 2.0× | 2.4× | 2.7× | 3.1× |
| CAL-SD | 92.6 ±0.1 | **94.0** ±0.2 | 94.5 ±0.1 | 94.8 ±0.2 | **94.9** ±0.1 | 1.4 × | 2.0× | 2.4× | 2.7× | 3.1× |
| CAL-SDS2 | 92.6 ±0.1 | **94.0** ±0.2 | **94.6** ±0.2 | **94.9** ±0.1 | **94.9** ±0.1 | 1.1× | 1.5× | 1.7× | 1.9× | 2.1× |
| AL w/ WS | **92.7** ±0.3 | 93.8 ±0.2 | 94.4 ±0.1 | 94.6 ±0.1 | 94.4 ±0.2 | 1.1× | 1.4× | 1.5× | 1.5× | 1.5× |
| AL | 92.6 ±0.3 | 93.8 ±0.0 | 94.4 ±0.1 | **94.9** ±0.2 | **94.9** ±0.1 | 1.0× | 1.0× | 1.0× | 1.0× | 1.0× |

Table 1: FMNIST Results

| Method | Test Accuracy (%) | | | | | Factor Speedup | | | | |
|---|---|---|---|---|---|---|---|---|---|---|
| | 10% | 20% | 30% | 40% | 50% | 10% | 20% | 30% | 40% | 50% |
| CAL-ER | 82.1 ±0.5 | 89.9 ±0.3 | 92.4 ±0.1 | 93.5 ±0.1 | 93.7 ±0.3 | 1.6× | 2.8× | 4.0× | 5.3× | 6.5× |
| CAL-MIR | 82.4 ±0.4 | 89.5 ±0.3 | 92.6 ±0.3 | 93.6 ±0.1 | 93.8 ±0.2 | 0.7 × | 1.0× | 1.4× | 1.8× | 2.2× |
| CAL-DER | **83.5** ±0.1 | 90.0 ±0.4 | 92.3 ±0.1 | 93.1 ±0.2 | 93.4 ±0.1 | 1.4× | 2.3× | 3.2× | 4.2× | 5.2× |
| CAL-SD | 83.0 ±0.0 | 90.0 ±0.4 | 92.7 ±0.2 | 93.3 ±0.3 | 93.9 ±0.3 | 1.4× | 2.2× | 3.2× | 4.1× | 5.1× |
| CAL-SDS2 | 82.5 ±0.1 | **90.1** ±0.2 | **92.9** ±0.4 | **94.0** ±0.2 | **94.4** ±0.1 | 1.1× | 1.6× | 2.1× | 2.7× | 3.4× |
| AL w/ WS | 81.9 ±0.4 | 89.6 ±0.5 | 92.4 ±0.2 | 93.5 ±0.1 | 94.1 ±0.1 | 1.2× | 1.1× | 1.1× | 1.1× | 1.1× |
| AL | 82.0 ±0.3 | 89.1 ±0.2 | 92.1 ±0.4 | 93.5 ±0.3 | 93.8 ±0.2 | 1.0× | 1.0× | 1.0× | 1.0× | 1.0× |

Table 2: CIFAR-10 Results

| | Test Accuracy (%) | | | | | Factor Speedup | | | | |
|---|---|---|---|---|---|---|---|---|---|---|
| **Method** | **10%** | **15%** | **20%** | **25%** | **30%** | **10%** | **15%** | **20%** | **25%** | **30%** |
| CAL-ER | $56.8_{\pm 10.1}$ | $62.6_{\pm 2.9}$ | $64.3_{\pm 3.9}$ | $63.8_{\pm 8.1}$ | $62.6_{\pm 3.0}$ | 1.5× | 2.3× | 2.8× | 3.3× | 3.9× |
| CAL-MIR | $60.8_{\pm 4.5}$ | $66.0_{\pm 3.8}$ | $64.3_{\pm 8.6}$ | $68.3_{\pm 2.1}$ | $69.5_{\pm 1.7}$ | 1.1× | 1.3× | 1.3× | 1.5× | 1.7× |
| CAL-DER | $65.5_{\pm 3.7}$ | $69.1_{\pm 0.8}$ | $\mathbf{69.9}_{\pm 0.3}$ | $70.1_{\pm 0.8}$ | $\mathbf{71.9}_{\pm 0.5}$ | 1.3× | 1.9× | 2.2× | 2.6× | 3.0× |
| CAL-SD | $62.9_{\pm 3.2}$ | $68.5_{\pm 0.5}$ | $69.3_{\pm 0.7}$ | $\mathbf{70.8}_{\pm 0.6}$ | $70.7_{\pm 1.3}$ | 1.3× | 1.9× | 2.2× | 2.6× | 3.0× |
| CAL-SDS2 | $61.3_{\pm 10.5}$ | $69.1_{\pm 2.5}$ | $69.4_{\pm 1.7}$ | $70.2_{\pm 0.8}$ | $70.7_{\pm 1.2}$ | 1.1× | 1.5× | 1.7× | 2.0× | 2.2× |
| AL w/ WS | $66.0_{\pm 0.9}$ | $65.6_{\pm 0.4}$ | $69.4_{\pm 0.9}$ | $69.7_{\pm 0.7}$ | $70.7_{\pm 0.4}$ | 1.1× | 1.4× | 1.6× | 2.1× | 2.2× |
| AL | $\mathbf{66.2}_{\pm 3.4}$ | $\mathbf{70.2}_{\pm 0.6}$ | $68.7_{\pm 2.5}$ | $69.4_{\pm 3.2}$ | $71.4_{\pm 1.2}$ | 1.0× | 1.0× | 1.0× | 1.0× | 1.0× |

Table 3: MedMNIST Results

| | Test Accuracy (%) | | | | | Factor Speedup | | | | |
|---|---|---|---|---|---|---|---|---|---|---|
| **Method** | **10%** | **20%** | **30%** | **40%** | **50%** | **10%** | **20%** | **30%** | **40%** | **50%** |
| CAL-ER | $90.7_{\pm 3.1}$ | $92.4_{\pm 1.2}$ | $93.5_{\pm 0.1}$ | $93.7_{\pm 0.2}$ | $93.8_{\pm 0.2}$ | 1.5x | 3.5x | 3.3x | 4.1x | 4.9x |
| CAL-MIR | $92.0_{\pm 0.9}$ | $92.9_{\pm 0.1}$ | $93.3_{\pm 0.3}$ | $93.7_{\pm 0.1}$ | $93.7_{\pm 0.2}$ | 0.9x | 1.3x | 1.8x | 2.3x | 2.7x |
| CAL-DER | $\mathbf{92.9}_{\pm 0.3}$ | $\mathbf{94.1}_{\pm 0.3}$ | $\mathbf{94.0}_{\pm 0.7}$ | $93.6_{\pm 0.8}$ | $\mathbf{94.2}_{\pm 0.3}$ | 1.5x | 2.4x | 3.2x | 4.0x | 4.7x |
| CAL-SD | $92.1_{\pm 0.3}$ | $92.6_{\pm 0.4}$ | $93.7_{\pm 0.1}$ | $\mathbf{93.8}_{\pm 0.1}$ | $94.1_{\pm 0.1}$ | 1.5x | 2.4x | 3.2x | 4.0x | 4.7x |
| CAL-SDS2 | $92.6_{\pm 0.3}$ | $93.2_{\pm 0.1}$ | $93.6_{\pm 0.1}$ | $\mathbf{93.8}_{\pm 0.4}$ | $94.1_{\pm 0.0}$ | 1.2x | 1.9x | 2.5x | 3.1x | 3.7x |
| AL w/ WS | $92.6_{\pm 0.5}$ | $93.0_{\pm 0.2}$ | $93.0_{\pm 0.1}$ | $93.2_{\pm 0.3}$ | $93.1_{\pm 0.1}$ | 1.0x | 1.0x | 1.0x | 1.0x | 1.0x |
| AL | $92.8_{\pm 0.2}$ | $93.1_{\pm 0.7}$ | $93.3_{\pm 1.1}$ | $\mathbf{93.8}_{\pm 0.5}$ | $94.1_{\pm 0.2}$ | 1.0x | 1.0x | 1.0x | 1.0x | 1.0x |

Table 4: Amazon Polarity Results

| | Test Accuracy (%) | | | | | Factor Speedup | | | | |
|---|---|---|---|---|---|---|---|---|---|---|
| **Method** | **2.9%** | **5.7%** | **8.6%** | **11.4%** | **14.3%** | **2.9%** | **5.7%** | **8.6%** | **11.4%** | **14.3%** |
| CAL-ER | $73.2_{\pm 1.7}$ | $75.4_{\pm 0.8}$ | $76.4_{\pm 1.0}$ | $77.0_{\pm 2.0}$ | $77.3_{\pm 1.4}$ | 1.7x | 2.8x | 3.9x | 5.0x | 6.1x |
| CAL-MIR | $\mathbf{75.1}_{\pm 0.2}$ | $75.5_{\pm 1.2}$ | $76.6_{\pm 1.0}$ | $76.5_{\pm 0.4}$ | $77.0_{\pm 0.3}$ | 0.8x | 1.2x | 1.6x | 2.0x | 2.4x |
| CAL-DER | $71.5_{\pm 2.7}$ | $74.9_{\pm 3.2}$ | $75.7_{\pm 1.5}$ | $76.9_{\pm 1.6}$ | $78.3_{\pm 0.8}$ | 1.7x | 2.7x | 3.7x | 4.8x | 5.8x |
| CAL-SD | $73.6_{\pm 1.9}$ | $74.9_{\pm 1.1}$ | $\mathbf{77.7}_{\pm 1.3}$ | $76.4_{\pm 0.3}$ | $78.0_{\pm 0.9}$ | 1.7x | 2.7x | 3.7x | 4.8x | 5.8x |
| CAL-SDS2 | $74.7_{\pm 2.8}$ | $75.5_{\pm 1.0}$ | $77.2_{\pm 0.9}$ | $\mathbf{78.1}_{\pm 0.8}$ | $\mathbf{79.2}_{\pm 0.5}$ | 1.4x | 2.1x | 2.9x | 3.7x | 4.5x |
| AL w/ WS | $74.6_{\pm 0.7}$ | $\mathbf{76.1}_{\pm 0.4}$ | $76.3_{\pm 1.0}$ | $76.3_{\pm 1.5}$ | $77.2_{\pm 0.9}$ | 1.2x | 1.7x | 1.6x | 1.8x | 1.8x |
| AL | $73.9_{\pm 2.9}$ | $75.5_{\pm 0.5}$ | $76.6_{\pm 2.0}$ | $76.3_{\pm 0.9}$ | $77.3_{\pm 1.6}$ | 1.0x | 1.0x | 1.0x | 1.0x | 1.0x |

Table 5: COLA Results.

| | Test Accuracy (%) | | | | | Factor Speedup | | | | |
|---|---|---|---|---|---|---|---|---|---|---|
| **Method** | **10%** | **15%** | **20%** | **25%** | **30%** | **10%** | **15%** | **20%** | **25%** | **30%** |
| CAL-ER | $86.3_{\pm 0.1}$ | $88.3_{\pm 0.1}$ | $89.7_{\pm 0.3}$ | $90.6_{\pm 0.2}$ | $91.0_{\pm 0.1}$ | 1.5× | 2.0× | 2.4× | 2.9× | 3.4× |
| CAL-MIR | $86.3_{\pm 0.1}$ | $88.3_{\pm 0.1}$ | $89.7_{\pm 0.2}$ | $90.5_{\pm 0.2}$ | $90.9_{\pm 0.2}$ | 1.2× | 1.6× | 1.9× | 2.2× | 2.6× |
| CAL-DER | $\mathbf{86.9}_{\pm 0.3}$ | $\mathbf{88.8}_{\pm 0.3}$ | $\mathbf{89.9}_{\pm 0.3}$ | $\mathbf{90.7}_{\pm 0.2}$ | $\mathbf{91.2}_{\pm 0.1}$ | 1.4× | 1.9× | 2.3× | 2.8× | 3.3× |
| CAL-SD | $86.3_{\pm 0.1}$ | $88.3_{\pm 0.1}$ | $89.5_{\pm 0.2}$ | $90.3_{\pm 0.2}$ | $90.8_{\pm 0.2}$ | 1.4× | 1.9× | 2.3× | 2.8× | 3.3× |
| CAL-SDS2 | $86.3_{\pm 0.1}$ | $88.2_{\pm 0.1}$ | $89.2_{\pm 0.3}$ | $90.1_{\pm 0.2}$ | $90.6_{\pm 0.1}$ | 1.4× | 1.9× | 2.3× | 2.8× | 3.3× |
| AL w/ WS | $86.3_{\pm 0.1}$ | $88.3_{\pm 0.1}$ | $88.4_{\pm 0.8}$ | $88.2_{\pm 0.8}$ | $88.1_{\pm 0.8}$ | 1.0× | 1.0× | 1.2× | 1.6× | 1.9× |
| AL | $83.6_{\pm 1.0}$ | $87.0_{\pm 0.3}$ | $88.6_{\pm 0.1}$ | $89.5_{\pm 0.2}$ | $89.9_{\pm 0.3}$ | 1.0× | 1.0× | 1.0× | 1.0× | 1.0× |

Table 6: Single-Cell Cell-Type Identity Classification Results

## A.2 HYPERPARAMETERS

For every dataset and every CAL/AL strategy, learning rate ($lr$) and batch size ($m$) are chosen based on whichever setting achieves highest performance on standard AL. For all CAL methods, replay size $m' \in \{m, 2m\}$ (used in all CAL methods), $\alpha \in \{0.1, 0.25, 0.5, 0.75\}$ (used in CAL-DER, CAL-SD,

and CAL-SDS2), $\beta \in \{0.75, 1\}$ (used in CAL-DER), $\sigma \in \{0.1, 1\}$ (used in CAL-SDS2), and $\lambda \in \{0.1, 1, 10\}$ (used in CAL-SDS2). $C$ is the hyperparameter used in CAL-MIR and CAL-SDS2 to subsample the history before finding the $m'$ samples to replay, but this parameter is not tuned for any of the presented results. We list the specific set of hyperparameters we use for all the main experimental results in this section.

### A.2.1   FMNIST

All experiments for FMNIST used a ResNet-18 with an SGD optimizer, with learning rate of 0.01 and batch size of 64. For all the CAL methods, we fix $m' = 128$. A NVIDIA GeForce RTX 1080 GPU was used to run all the reported experiments.

**CAL-MIR**   $C = 256$

**CAL-DER**   $\alpha = .1, \beta = 1$

**CAL-SD**   $\alpha = .25$

**CAL-SDS2**   $C = 256, \alpha = .25, \sigma = 0.1, \lambda = 1$

### A.2.2   CIFAR-10

All experiments for CIFAR-10 used a ResNet-18 with an SGD optimizer, with learning rate of 0.02 and a batch size of 20. For all the CAL methods, we fix $m' = 40$. Training is done on an NVIDIA GeForce RTX 2080.

**CAL-MIR**   $C = 100$

**CAL-DER**   $\alpha = .1, \beta = 1$

**CAL-SD**   $\alpha = .25$

**CAL-SDS2**   $C = 100, \alpha = .25, \sigma = 0.1, \lambda = 0.1$

### A.2.3   MEDMNIST

All experiments for MedMNIST used a ResNet-18 with an Adam optimizer, with learning rate of 0.001 and a batch size of 128. For all CAL methods, we fix $m' = 128$. All reported models were trained on an NVIDIA GeForce RTX 2080.

**CAL-MIR**   $C = 270, m' = 128$

**CAL-DER**   $m' = 128, \alpha = .1, \beta = 1$

**CAL-SD**   $m' = 128, \alpha = .5$

**CAL-SDS2**   $C = 270, m' = 128, \alpha = .5, \sigma = 0.1, \lambda = 10$

### A.2.4   AMAZON POLARITY REVIEW

Throughout our experiments, we sample 2M sentences, and use them as the total training set instead. We use Adam optimizer with standard parameters with learening rate of 0.001 and a batch size 128. For all the CAL methods, we fix $m' = 128$. All reported models were trained on an NVIDIA GeForce 1080 Ti.

**CAL-MIR**   $C = 256,$

**CAL-DER**   $\alpha = .25, \beta = 0.75$

**CAL-SD**   $\alpha = .5$

**CAL-SDS2**   $C = 256, \alpha = .75, \sigma = 1, \lambda = 1$

### A.2.5   COLA

For all of our experiments we use Huggingface's transformer library Wolf et al. (2020) and use a maximum sentence length of 100. We use Adam optimizer and a learning rate of $5 \cdot 10^{-5}$, use a batch size of 25 and $m' = 25$. Models were trained on a single NVIDIA GeForce 1080 Ti.

**CAL-MIR**   $C = 50$

**CAL-DER**   $\alpha = 0.25, \beta = 0.75$

**CAL-SDS**   $\alpha = 0.75, \beta = 0.25$

**CAL-SDS2**   $C = 50, \alpha = 0.5, \beta = 0.5, \sigma = 1, \lambda = 1.$

### A.2.6   SINGLE-CELL CELL-TYPE IDENTITY CLASSIFICATION

All experiments use SGD optimizer with standard parameters with learning rate of 0.001 and a batch size 128. For all the CAL methods, we fix $m' = 128$. Training is done on an NVIDIA A100-PCIE-40GB.

**CAL-MIR**   $C = 200,$

**CAL-DER**   $\alpha = .1, \beta = 1$

**CAL-SD**   $\alpha = 1$

**CAL-SDS2**   $C = 100, \alpha = .25, \sigma = 0.1, \lambda = 1$

## B   RESULTS FOR ADDITIONAL ACTIVE LEARNING STRATEGIES

In this section, we demonstrate that CAL methods are able to accelerate AL strategies other than entropy sampling without incurring any significant performance drops. We test multiple AL strategies on and FMNIST Xiao et al. (2017) and CIFAR-10 Krizhevsky (2009). Note that the speedups are approximately the same as the ones reported in Section A since the training time is generally independent of the selected AL strategy.

### B.1   OVERVIEW OF STRATEGIES

**Margin Score Sampling**   This strategy is another form of uncertainty sampling Settles (2009) as described in the main paper. Instead of the entropy of $f(x; \theta)$, the margin score is used as the entropy score i.e. $h(x) \triangleq 1 - (f(x; \theta)_i - f(x; \theta)_j)$ where $i$ and $j$ are the indices corresponding to the highest and second highest values of $f(x; \theta)$ respectively.

**FASS**   FASS Wei et al. (2015) is a two-staged selection method that uses both uncertainty sampling and submodular maximization. Initially, a set of samples $\mathcal{A}$ of cardinality $c * b_t$ is chosen from $\mathcal{U}$ using uncertainty sampling, where $c > 1$ is a tuneable hyperparameter. Next, $U_t$ is constructed by greedily selecting samples that maximize a submodular set function $G : 2^{\mathcal{A}} \to \mathbb{R}_+$ defined on ground set $\mathcal{A}$. Entropy is once again used as the uncertainty metric for the initial stage. For the second stage, $G$ is defined to be the facility location function Wei et al. (2015) expressed below:

$$G(\mathcal{S}) = \sum_{x_i \in \mathcal{A}} \max_{x_j \in \mathcal{S}} w_{ij}, \tag{7}$$

where $\mathcal{S} \subseteq \mathcal{A}$ and $w_{ij}$ is a similarity score between samples $x_i$ and $x_j$. In our experiments, $w_{ij} = \exp\left(-\|z_i - z_j\|^2/2\sigma^2\right)$ where $z_i$ is the penultimate layer representation of model $f$ for $x_i$ and $\sigma$ is a hyperparameter.

**GLISTER**  GLISTER Killamsetty et al. (2021) solves a bi-level optimization problem in order to select samples to label. Specifically, GLISTER solves

$$\underset{\mathcal{S} \subseteq \mathcal{U}_t, |\mathcal{S}| \le b_t}{\arg\max} \; LL_V(\underset{\theta}{\arg\max} \, LL_T(\theta, \mathcal{S}), \mathcal{V}) \tag{8}$$

where $LL_V$ is the log-likelihood on the validation set $\mathcal{V}$, and $LL_T$ is the log-likelihood on the subset $\mathcal{S}$.

## B.2 RESULTS

| Method | Test Accuracy (%) | | | | |
|---|---|---|---|---|---|
| | **10%** | **15%** | **20%** | **25%** | **30%** |
| CAL-ER | **92.8** $\pm$ 0.1 | **94.1** $\pm$ 0.1 | 94.8 $\pm$ 0.1 | **95.1** $\pm$ 0.3 | **95.2** $\pm$ 0.2 |
| CAL-MIR | 92.6 $\pm$ 0.2 | **94.1** $\pm$ 0.4 | **94.9** $\pm$ 0.2 | 95.0 $\pm$ 0.2 | **95.2** $\pm$ 0.2 |
| CAL-DER | 91.8 $\pm$ 0.5 | 93.1 $\pm$ 0.1 | 94.3 $\pm$ 0.3 | 94.6 $\pm$ 0.1 | 94.8 $\pm$ 0.2 |
| CAL-SD | 92.5 $\pm$ 0.1 | 93.8 $\pm$ 0.1 | 94.8 $\pm$ 0.0 | **95.1** $\pm$ 0.2 | **95.2** $\pm$ 0.0 |
| CAL-SDS2 | 87.8 $\pm$ 1.1 | 93.4 $\pm$ 0.1 | 94.6 $\pm$ 0.1 | 95.0 $\pm$ 0.2 | **95.2** $\pm$ 0.1 |
| AL w/ WS | **92.8** $\pm$ 0.0 | 94.0 $\pm$ 0.3 | 94.6 $\pm$ 0.1 | 94.8 $\pm$ 0.1 | 95.0 $\pm$ 0.2 |
| AL | 92.7 $\pm$ 0.1 | **94.1** $\pm$ 0.3 | **94.9** $\pm$ 0.1 | 95.0 $\pm$ 0.2 | **95.2** $\pm$ 0.1 |

Table 7: FMNIST with Margin Score Sampling

| Method | Test Accuracy (%) | | | | |
|---|---|---|---|---|---|
| | **10%** | **20%** | **30%** | **40%** | **50%** |
| CAL-ER | 81.5 $\pm$ 0.1 | 89.3 $\pm$ 0.1 | 92.2 $\pm$ 0.2 | 93.4 $\pm$ 0.1 | 93.8 $\pm$ 0.0 |
| CAL-MIR | 81.9 $\pm$ 0.1 | 89.6 $\pm$ 0.2 | 92.2 $\pm$ 0.4 | 93.6 $\pm$ 0.0 | 94.0 $\pm$ 0.2 |
| CAL-DER | 83.0 $\pm$ 0.2 | 89.5 $\pm$ 0.2 | 92.2 $\pm$ 0.2 | 93.2 $\pm$ 0.2 | 93.6 $\pm$ 0.0 |
| CAL-SD | 82.6 $\pm$ 0.4 | 89.9 $\pm$ 0.4 | 92.4 $\pm$ 0.2 | 93.5 $\pm$ 0.1 | 93.8 $\pm$ 0.2 |
| CAL-SDS2 | 82.5 $\pm$ 0.2 | 90.2 $\pm$ 0.2 | 92.5 $\pm$ 0.2 | **93.8** $\pm$ 0.2 | **94.1** $\pm$ 0.1 |
| AL w/ WS | **83.1** $\pm$ 0.1 | **90.3** $\pm$ 0.3 | **93.0** $\pm$ 0.2 | 93.5 $\pm$ 0.3 | 93.6 $\pm$ 0.2 |
| AL | 75.1 $\pm$ 1.2 | 87.1 $\pm$ 1.0 | 90.2 $\pm$ 0.5 | 92.0 $\pm$ 0.0 | 92.8 $\pm$ 0.5 |

Table 8: CIFAR-10 with Margin Score Sampling

| | Test Accuracy (%) | | | | |
|---|---|---|---|---|---|
| Method | 10% | 15% | 20% | 25% | 30% |
| CAL-ER | $92.6_{\pm 0.1}$ | $\mathbf{93.9}_{\pm 0.2}$ | $94.6_{\pm 0.2}$ | $\mathbf{95.0}_{\pm 0.1}$ | $94.9_{\pm 0.0}$ |
| CAL-MIR | $92.5_{\pm 0.1}$ | $93.8_{\pm 0.3}$ | $94.6_{\pm 0.1}$ | $94.8_{\pm 0.1}$ | $94.9_{\pm 0.2}$ |
| CAL-DER | $92.7_{\pm 0.1}$ | $93.8_{\pm 0.1}$ | $94.5_{\pm 0.1}$ | $94.7_{\pm 0.1}$ | $\mathbf{95.0}_{\pm 0.2}$ |
| CAL-SD | $\mathbf{92.8}_{\pm 0.1}$ | $\mathbf{93.9}_{\pm 0.1}$ | $\mathbf{94.7}_{\pm 0.1}$ | $94.8_{\pm 0.3}$ | $94.9_{\pm 0.1}$ |
| CAL-SDS2 | $\mathbf{92.8}_{\pm 0.0}$ | $93.8_{\pm 0.2}$ | $94.5_{\pm 0.1}$ | $94.8_{\pm 0.2}$ | $94.9_{\pm 0.1}$ |
| AL w/ WS | $92.5_{\pm 0.1}$ | $93.8_{\pm 0.3}$ | $94.0_{\pm 0.2}$ | $94.3_{\pm 0.2}$ | $94.3_{\pm 0.0}$ |
| AL | $92.7_{\pm 0.4}$ | $\mathbf{93.9}_{\pm 0.1}$ | $94.5_{\pm 0.1}$ | $94.7_{\pm 0.3}$ | $94.8_{\pm 0.1}$ |

Table 9: FMNIST with FASS

| | Test Accuracy (%) | | | | |
|---|---|---|---|---|---|
| Method | 10% | 20% | 30% | 40% | 50% |
| CAL-ER | $82.2_{\pm 0.2}$ | $89.8_{\pm 0.2}$ | $92.5_{\pm 0.2}$ | $93.4_{\pm 0.4}$ | $93.7_{\pm 0.2}$ |
| CAL-MIR | $82.2_{\pm 0.3}$ | $89.4_{\pm 0.2}$ | $92.3_{\pm 0.1}$ | $93.4_{\pm 0.0}$ | $93.5_{\pm 0.1}$ |
| CAL-DER | $\mathbf{83.1}_{\pm 0.3}$ | $89.7_{\pm 0.2}$ | $91.9_{\pm 0.1}$ | $93.1_{\pm 0.2}$ | $93.5_{\pm 0.1}$ |
| CAL-SD | $83.0_{\pm 0.3}$ | $90.0_{\pm 0.3}$ | $92.5_{\pm 0.1}$ | $93.5_{\pm 0.1}$ | $\mathbf{94.0}_{\pm 0.1}$ |
| CAL-SDS2 | $83.0_{\pm 0.1}$ | $90.1_{\pm 0.1}$ | $92.7_{\pm 0.2}$ | $93.5_{\pm 0.2}$ | $\mathbf{94.0}_{\pm 0.0}$ |
| AL w/ WS | $82.8_{\pm 0.4}$ | $\mathbf{90.3}_{\pm 0.1}$ | $\mathbf{92.8}_{\pm 0.2}$ | $\mathbf{93.6}_{\pm 0.1}$ | $93.7_{\pm 0.3}$ |
| AL | $72.5_{\pm 2.0}$ | $86.6_{\pm 0.4}$ | $90.1_{\pm 0.4}$ | $91.7_{\pm 0.2}$ | $92.9_{\pm 0.2}$ |

Table 10: CIFAR-10 with FASS

| | Test Accuracy (%) | | | | |
|---|---|---|---|---|---|
| Method | 10% | 15% | 20% | 25% | 30% |
| CAL-ER | $92.6_{\pm 0.0}$ | $\mathbf{93.9}_{\pm 0.2}$ | $94.3_{\pm 0.1}$ | $\mathbf{94.7}_{\pm 0.1}$ | $94.7_{\pm 0.2}$ |
| CAL-MIR | $92.5_{\pm 0.0}$ | $\mathbf{93.9}_{\pm 0.4}$ | $94.3_{\pm 0.2}$ | $94.4_{\pm 0.2}$ | $94.6_{\pm 0.1}$ |
| CAL-DER | $92.7_{\pm 0.1}$ | $\mathbf{93.9}_{\pm 0.2}$ | $94.3_{\pm 0.3}$ | $\mathbf{94.7}_{\pm 0.2}$ | $94.9_{\pm 0.3}$ |
| CAL-SD | $92.6_{\pm 0.1}$ | $93.8_{\pm 0.1}$ | $\mathbf{94.4}_{\pm 0.3}$ | $94.6_{\pm 0.1}$ | $94.7_{\pm 0.1}$ |
| CAL-SDS2 | $92.6_{\pm 0.1}$ | $\mathbf{93.9}_{\pm 0.2}$ | $\mathbf{94.4}_{\pm 0.2}$ | $94.6_{\pm 0.3}$ | $94.7_{\pm 0.2}$ |
| AL w/ WS | $92.5_{\pm 0.1}$ | $93.6_{\pm 0.1}$ | $93.9_{\pm 0.1}$ | $94.1_{\pm 0.1}$ | $94.3_{\pm 0.1}$ |
| AL | $92.5_{\pm 0.2}$ | $93.8_{\pm 0.1}$ | $94.2_{\pm 0.1}$ | $94.6_{\pm 0.2}$ | $94.7_{\pm 0.2}$ |

Table 11: FMNIST with GLISTER

| | Test Accuracy (%) | | | | |
|---|---|---|---|---|---|
| Method | 10% | 20% | 30% | 40% | 50% |
| CAL-ER | $81.7_{\pm 0.3}$ | $89.2_{\pm 0.2}$ | $91.9_{\pm 0.2}$ | $93.0_{\pm 0.1}$ | $93.3_{\pm 0.1}$ |
| CAL-MIR | $81.6_{\pm 0.3}$ | $89.3_{\pm 0.4}$ | $91.7_{\pm 0.2}$ | $92.9_{\pm 0.1}$ | $93.5_{\pm 0.2}$ |
| CAL-DER | $\mathbf{82.8}_{\pm 0.4}$ | $89.5_{\pm 0.4}$ | $91.7_{\pm 0.4}$ | $92.8_{\pm 0.6}$ | $93.1_{\pm 0.2}$ |
| CAL-SD | $82.5_{\pm 0.3}$ | $\mathbf{89.6}_{\pm 0.2}$ | $92.1_{\pm 0.2}$ | $93.1_{\pm 0.2}$ | $93.8_{\pm 0.1}$ |
| CAL-SDS2 | $81.4_{\pm 0.4}$ | $89.1_{\pm 0.2}$ | $92.1_{\pm 0.2}$ | $93.2_{\pm 0.3}$ | $\mathbf{93.9}_{\pm 0.1}$ |
| AL w/ WS | $81.7_{\pm 0.4}$ | $89.3_{\pm 0.4}$ | $92.1_{\pm 0.3}$ | $93.0_{\pm 0.1}$ | $93.3_{\pm 0.4}$ |
| AL | $81.0_{\pm 0.6}$ | $88.5_{\pm 0.5}$ | $91.5_{\pm 0.3}$ | $93.0_{\pm 0.2}$ | $93.4_{\pm 0.3}$ |

Table 12: CIFAR-10 with GLISTER

# C    ADDITIONAL DETAILS ON SINGLE-CELL CELL-TYPE IDENTITY CLASSIFICATION DATASET

The human cell landscape (HCL) dataset consists of scRNA-seq data for 562,977 cells across 63 cell types represented in 56 human tissues. Each cell type may be present in multiple tissues. The cell type classes are highly imbalanced, with the rarest cell type, human embryonic stem cell, accounting for 0.00006 % of the total dataset and the most common, fibroblast, accounting for 0.06%. The raw data is first normalized for library-size and scaled to 10000 reads in total, followed by log-transformation. We visualize the dataset using UMAP 9.

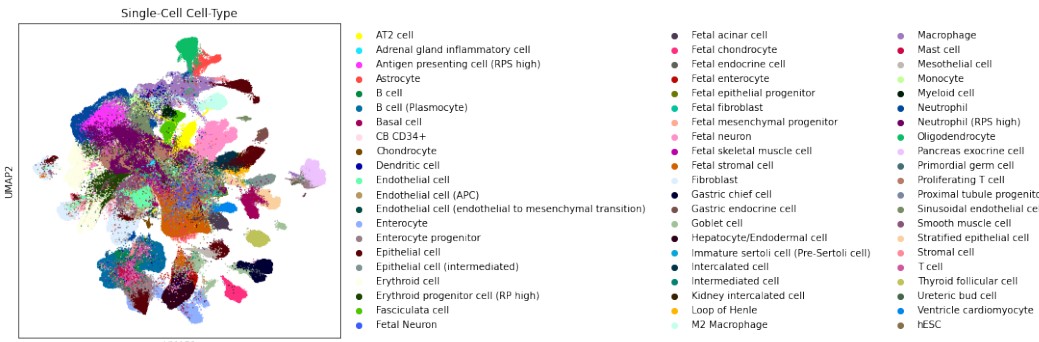

Figure 9: UMAP embedding of single cells in HCL annotated by their cell type.

