# OpenReview forum: "Continual Active Learning"
_ICLR.cc/2023/Conference — Submitted to ICLR 2023_

### Official Review · Reviewer_nCXE · 2022-10-22

**Confidence:** 4
**Clarity, Quality, Novelty And Reproducibility:** Have met the standards.
**Correctness:** 3
**Technical Novelty And Significance:** 2
**Empirical Novelty And Significance:** 2
**Recommendation:** 3

**Strength And Weaknesses:**

The idea of this paper is well-received.

My main concern is listed as follows.
1. the novelty. It is yet to be answered if the performance gain is truly brought by the active learning framework, or the promotion from the classifiers.
2. this paper may seem a bit incremental to me where a lot of existing techniques were combined into a whole.
3. following 2, the integration of the existing continual learning techniques are generally loose. In other words, how can we attain better querying policy via the integration of continual learning remains untapped.
4. the establishment of the new setting needs further explanation.

The other more detailed weaknesses are:

1. With all due respect, I may disagree with the author on the simple statement, "active learning can be viewed as a continual learning problem". While indeed that the active learning and continual learning seem to have something in common (i.e. new sample being added into the pool at every cycle), crucially a more vital stage of the active learning is the querying policy of attaining new batch of unlabeled data that are gauged to bring maximal performance gain. The paper didn't explain this aspect adequately.

2. Section 3 is occupied by the existing settings or techniques. It seems to me that the section 3.1 and 3.2 fit the background interpretation while section 3.3 describes the existing continual learning techniques. These, however, seems to very much orthogonal to the active learning.

3. Following 2, the major methodological innovation lies in the proposition of CAL-SD and CAL-SDS2. It generally lacks essential details that can correlate active learning and continual learning. For example, this paper emphasize a "stability-plasticity" problem in CAL-DS, but I don't see a clear connection with the active learning. The sub-modular sampling in CAL-SDS2 also seems orthogonal to active learning. To summarize, this paper seems to simply utilize the continual learning to an active learning setup incrementally.

4. The empirical analysis in section 4.2 is a bit hard for me to understand, with a surprisingly short piece of analysis. I suggest the authors to focus on CAL-SD and CAL-SDS2 and offer an ablation analysis accordingly, such as the impact of $\mathcal{L}_c$ and $\mathcal{L}_{\rm replay}$.

5. A major problem for active learning is the costly querying process, which is not shown in this paper because the authors only apply continual learning to uncertainty sampling strategies, while there are many existing. I suggest that the authors should consider other types of baselines.

**Summary Of The Paper:**

This paper introduces a continual learning technique for the active learning. In particular, a set of the continual learning techniques are adopted and integrated into each model training procedure at each cycle of active learning. The proposed CAL regards the newly-queried sample at each round a new task as in the continual learning setting, then apply continual learning techniques combined with warm-starting to active learning. As a result, CAL accelerates the active learning process by improving computational efficiency at the model training stage in active learning. The contribution is demonstrated with extensive experiments in benchmarks across different domains (NLP, vision, medical imaging, and computational biology).

**Summary Of The Review:**

Based on the above comments, I recommend a reject on this paper.

This paper should significantly strengthen the integration of the active learning and continual learning, rather than simply superficially coupling both.

---

> ### Author Response · Authors · 2022-11-19
> **We thank the reviewer for their time and feedback, and are glad that the reviewer finds the work well-motivated. Below we provide our responses to the reviewer’s concerns:**
>
> We thank the reviewer for their time and feedback and are glad that the reviewer finds the work well-motivated. Below we provide our responses to the reviewer’s concerns:
>
> - The vital stage of active learning is the querying policy of attaining new batch of unlabeled data that are gauged to bring maximal performance gain. Therefore, it is inaccurate to state that active learning is a form of continual learning.
>
> The distinctive feature of the continual learning (CL) paradigm is that the data presented to the model is sampled from non-stationary distribution [3]. Thus, there is nothing inherent in the definition of CL that precludes the task from being dependent on the model. Since the distribution of the labeled samples at a given active learning (AL) round is time variant (evidence of this is shown in Fig. 1), AL can in fact be viewed as a CL setting. This is what we refer to as CAL.
>
> - Section 3 is occupied by the existing settings or techniques
>
> We would like to point out that a background (algorithmically) of both AL and CL is needed to formulate CAL, therefore, it is not orthogonal to CAL. Furthermore, it is also important to talk about the existing CL algorithms and motivate CAL-SD and CAL-SDS2. Therefore, we kept the definitions of CL algorithms in the main paper.
>
> - “Following 2, the major methodological innovation lies in the proposition of CAL-SD and CAL-SDS2”
>
> We would like to point out that “stability-plasticity” is very important when talking about CAL. since it avoids re-training on all of the previously seen examples. A network is stable if it can effectively retain past information but cannot adapt to new tasks efficiently, whereas a network that is plastic can quickly learn new tasks but is prone to forgetting. With CAL, we would want to retain the previous knowledge (examples acquired during previous rounds) while at the same time learning new information (newly labeled examples). One thing to note is that as we proceed with labeling new samples with AL, the amount of new information added over time is less compared to the earlier rounds. Therefore In the context of CAL, we would like the model to be plastic during the early rounds and stable during the later rounds.
>
> - "The sub-modular sampling in CAL-SDS2 also seems orthogonal to active learning."
>
> Many replay-based methods use uniform random sampling from the memory buffer to fetch the data points. On the other hand, theoretical and empirical works on subset selection in various areas have shown superior performance of using a non-uniform, and in particular submodular maximization-based selection [4]. Therefore, this submodular part is done to improve the Continual learning side of CAL.
>
> -  “The empirical analysis in section 4.2 is a bit hard for me to understand..”
>
> The empirical analysis done in the mentioned section is inspired by [2] (Fig 3b.)  which used proxy models to select the examples (say 50% budget) to accelerate AL. We posit it is important to make sure that acceleration doesn’t bring harm to the metrics beyond accuracy. One such common metric is computing rank correlation based on the uncertainty of the held-out set, which we report in section 4.2.
>
> - “I suggest the authors focus on CAL-SD and CAL-SDS2 and offer an ablation analysis accordingly”
>
> CAL-SDS2 with ablation done on submodular selection is CAL-SD. On considering ablation studying the impact of $\mathcal{L}_c$ and $\mathcal{L}_{\rm replay}$ if we remove the replay part, then as discussed in section 3.3, we will suffer from catastrophic forgetting.
>
> - CAL does not address the cost of querying which is a major bottleneck of AL, and only reports speedups with uncertainty sampling as the acquisition function.
>
> The speedups reported only consider the training time of the model, and therefore the numbers will not change if the acquisition function is changed (we have added clarification of this in the revised version). We ensure that CAL preserves/improves the predictive performance of the model irrespective of AL strategy by running experiments on a few different acquisition functions in Appendix B.
>
> The reviewer is correct in pointing out that CAL does not mitigate the cost of querying, which is beyond the scope of the issue we seek to mitigate. However, the advantage of CAL is that it is complementary to existing approaches [1,2] that do accelerate the querying cost as we state in Section 2.

---

> > ### Author Response · Authors · 2022-11-19
> > **Adding References**
> >
> >
> > [1] Cody Coleman, Edward Chou, Sean Culatana, Peter Bailis, Alexander C. Berg, Roshan Sumbaly, Matei Zaharia, and I. Zeki Yalniz. Similarity search for efficient active learning and search of rare concepts. CoRR, abs/2007.00077, 2020a.
> >
> > [2] Cody Coleman, Christopher Yeh, Stephen Mussmann, Baharan Mirzasoleiman, Peter Bailis, Percy Liang, Jure Leskovec, and Matei Zaharia. Selection via proxy: Efficient data selection for deep learning. In International Conference on Learning Representations, 2020b.
> >
> > [3] Parisi et al. Continual Lifelong Learning with Neural Networks: a review, 2019
> >
> > [4] Jeff Bilmes 2022. Submodularity In Machine Learning and Artificial Intelligence
> >
> > *We once again thank the reviewer for their feedback and hope that the rebuttal clarifies the questions raised by them. We are happy to discuss if you need any further clarifications to increase the score and facilitate acceptance of the paper.*

---

### Official Review · Reviewer_wxxe · 2022-10-23

**Confidence:** 4
**Correctness:** 3
**Technical Novelty And Significance:** 4
**Empirical Novelty And Significance:** 4
**Recommendation:** 6

**Clarity, Quality, Novelty And Reproducibility:**

The writing is clear and easy to follow. The quality of the writing is good. That said, as mentioned before, a bigger focus needs to be put in the discussion of the results. Moreover, experimental details, like how the cross-validation of models was executed, and how many seeds were used in the experiments should go in the main paper. I appreciate that the authors detailed the hyperparameters in the appendix.
The work and its perspective on the connection between an AL and CL is novel and original.

**Strength And Weaknesses:**

### Strengths
1. The paper tackles an important, yet understudied, problem of active learning : the computation cost. Given the resources required for training every larger models, I agree with the authors, and believe that this cost should be addressed. This way, practitioners can compare the cost of labeling vs the cost of training models.
2. The authors make an interesting (and novel to the best of my knowledge) connection with Continual Learning. By doing so, they show that standard CL techniques can be applied in the AL setting with success.
3. The authors provide some experiments over different modalities and labelling budgets.

### Weaknesses
1. The paper is in dire need of a better discussion of the results. I say this for the following reason : The authors **show that using replay (CAL-ER) consistently outperforms regular Active Learning **. My understanding is that this finding directly contradicts the base experiment (figure 1) in (Ash & Adams) [1], where they found that doing full-replay from a warm-started model (CAL-ER), still underperforms a model trained from scratch on all the data (AL). In other words, I think the authors should provide insights (or at least discuss) how a CL model manages to outperform a model trained from scratch on all the data seen so far, which has long been seen as an upper bound in performance.
2. On Catastrophic forgetting in AL (fig 1). How are the models cross-validated at every round in CAL ? I don't quite agree with using the word "catastrophic forgetting" to describe the behavior in figure 1, since the test accuracy (in general) gets better. It seems more like the the model is overfitting on the training samples of each round.
3. The paper would greatly benefit from larger scale experiments. I think the question of compute efficiency is much more relevant in such settings. I understand the increased cost that's required to run these.

[1] On Warm-Starting Neural Network Training, https://arxiv.org/pdf/1910.08475.pdf, Neurips 2020

**Summary Of The Paper:**

The paper proposes to investigate the computational efficiency of Active Learning (AL). The authors argue that typical AL methods are computationally intensive, as models are retrained from scratch at every round. They propose to reframe the AL setting as a Continual Learning one, which allows them to leverage the vast literature on replay-based CL. The authors conduct a series of experiments testing a variety of existing and novel rehearsal based methods in their new Continual Active Learning setup. Interestingly, the authors show that using CL methods in Active Learning not only provides significant (2-6x) computational gains, but does so without incurring a loss in performance.

**Summary Of The Review:**

In summary, the authors address an important limitation in the field of Active Learning. I believe that with a more detailed analysis of the results, and a clear explanation as to why replay based methods can actually be more efficient and performant than their iid counterpart, would make this a strong paper.
For now, given that several key things are missing from the paper, I think this paper still needs work. Should the authors provide more insight and a clear response to the issue raised above, I believe the paper will be worthy of acceptance.


edit : after the rebuttal I have increased my score to a weak accept.

---

> ### Author Response · Authors · 2022-11-18
> **We thank the reviewer for their time and feedback. Below we provide our response to the reviewer’s primary concerns--**
>
> We thank the reviewer for their time and feedback. Below we provide our response to the reviewer’s primary concerns:
>
> - Why does CAL-ER consistently outperform standard AL? This observation is inconsistent with the results of Ash & Adams.
>
> We would first like to point out that the experiments in [1] do not consider the AL setting and only include experiments where the source/target datasets are unbiased estimates of the true distribution. Since the experimental setting of [1] is fundamentally different than ours, our results neither directly contradict or corroborate their results. A study by [2] studies the effect of warm starting in the context of AL and finds that there is ultimately no strong effect to warm-starting in later rounds.
>
> Furthermore, CAL methods not only warm-start the model between rounds but also bias learning towards newly labeled points. We hypothesize that generalization on the test set improves because 1) new information is learnt more efficiently by all CAL methods and 2) the addition of a distillation term in the loss (as done in CAL-DER, CAL-SD, and CAL-SDS2) helps regularize the model.
>
> - The phenomenon shown in Figure 1 is a symptom of overfitting rather than catastrophic forgetting.
>
> Catastrophic forgetting [1] is the term used to describe the tendency of neural networks to forget previously learned information upon learning new information. Thus, the precipitous drop in performance on tasks 1 through n-1 when upon learning on task n can be described as catastrophic forgetting irrespective of whether the model was overfitted to task n-1 or if the performance on the test set increases. Furthermore, the increase in the test set is minimal and the final performance is considerably lower than the numbers reported in Section 4.2. Even if it did increase it would not invalidate the claim that the model is exhibiting catastrophic forgetting.
>
> - The paper would benefit from larger-scale experiments
>
> The paper includes several experiments on large-scale datasets such as the Amazon Polarity dataset (consisting of $\sim$ 2M samples) and the Single-Cell Cell Type Identity Classification dataset ($\sim$600k samples). The set of datasets included in the experiments section is diverse in terms of their scales and modalities. Moreover, we explore modalities (medical imaging, NLP, biology) that are typically not used to benchmark active learning algorithms which we hope will be adopted by future work.
>
>
> *We once again thank the reviewer for their feedback and hope that the rebuttal clarifies the questions raised by them. We are happy to discuss if you need any further clarifications to increase the score and facilitate acceptance of the paper.*
>
> References:
> [1]  Robert M French. Catastrophic forgetting in connectionist networks. Trends in cognitive sciences

---

> > ### Comment · Reviewer_wxxe · 2022-11-24
> > **Response**
> >
> > Thank you for the concise rebuttal.
> >
> > *On the difference between the CAL setting and the one in Ash and Adams*
> > Thank you for the clarification. It is true that since the selected points are not selected at random the comparison does not hold.
> >
> > *On the definition of catastrophic forgetting*
> > Catastrophic forgetting is typically defined on performance over a held-out set of samples. Nevertheless, I understand the point you are making and I find it valid, irrespective of how it is termed.
> >
> > Lastly, by large-scale experiments I was referring more to the compute required than on the number of training samples. I feel strongly that the value of this type of analysis is to provide compute-efficient ways to solve the CAL problem, which is (in my opininion) much more interesting on complex large(r) scale problems.
> >
> >
> > Thank you.

---

### Official Review · Reviewer_HMvu · 2022-10-25

**Confidence:** 4
**Correctness:** 2
**Technical Novelty And Significance:** 3
**Empirical Novelty And Significance:** 2
**Recommendation:** 5

**Clarity, Quality, Novelty And Reproducibility:**

I find the paper Clarity problematic at some places, please see my comments above.
The novelty has a problematic aspect to it in the sense that the empirical results don't seem to support novelty of the newly proposed CAL criteria.
There is clearly some originality here but the empirical results dont seem to support its contribution.
I find the quality of the work satisfying.

**Strength And Weaknesses:**

Strengths
1.	The problem tackled in this work is important improving the running time and performance of Active  Learning techniques.
2.	The Authors show in their experiments that indeed it is important to do continual learning and that acceleration can be obtained in this manner.
3.	The paper is well organized
Weaknesses
1.	My main concern is that the empirical results show the simple CAL-ER is the actual winner in acceleration in most cases, and often even the trade-off with accuracy makes the favorable one. Well,…that means that there is no point in any of the ‘intelligent’ criteria, but just to have a diverse set (which can be obtained by the simple uniform sampling of CAL-ER). I would like to get the authors opinion on this observation, please.
2.	There are some parts that are unclear. For example,
a.	the setting of experiment in 3.3 isnt clear and confusing: its not clear if the model is fully trained first and with what data? What is the accuracy measured for? what does the statement ‘the model performs considerably worse on all of the tasks that is does on the test set? what are you measuring the performance on, if not the test set.
b.	what are “reasonable choices of m’ “?
3.	I don’t understand how the CAL-MIR selection of m’ is done. Is it done exhaustively over all the loss minimizer subsets? That prohibitive at some point, right?
4.	How is the selection of alpha and beta are done for (2) in the experiments?


**Summary Of The Paper:**

The paper suggests a set of selection criteria to training points in each AL step to avoid the so-called catastrophic forgetting which degrades model performance. A variety of heuristics is proposed some motivated by prior work and 2 new ones proposed by the authors (CAL-SD and CAL_SDS2).

**Summary Of The Review:**

To this end I feel the paper still needs to be refined, and the empirical test show a trend that reduces the novelty and significance of the proposed sampling criteria for CAL. Nevertheless, this is an important problem and I would like to authors to continue its pursuance.

---

> ### Author Response · Authors · 2022-11-18
> **We thank the reviewer for their time and feedback. We are glad that they found this problem to be well-motivated and the paper to be well-written.**
>
> We thank the reviewer for their time and feedback. We are glad that they found this problem to be well-motivated and the paper to be well-written. Below we provide our response to the reviewer’s primary concerns:
>
> - Since CAL-ER achieves the highest speedup, what is the purpose of using any other CAL method?
>
> CAL-ER achieves the highest speedup since random selection from the history is cheap and because there is no additional time/storage overhead from computing the distillation-based loss terms. However, CAL-ER also generally tends to underperform other CAL methods as it appears that distillation tends to significantly boost performance.
>
> In some settings, it may be the case that we are willing to tolerate some depreciation in performance to realize the speedup of CAL-ER. However, in other settings (i.e.medical imaging/computational biology/autonomous driving) performance degradation can have dire real-world implications. In this work, our goal is not to champion a single CAL method, but rather to propose an assortment of CAL techniques that all have different performance/speedup tradeoffs. Therefore, practitioners have the ability to determine which CAL method is most appropriate for their specific use case depending on their error tolerance.
>
> - Clarity issues: Clarification on the experiment shown in Sec 3.3
>
> The purpose of the experiment in Figure 1 is to show that models exhibit catastrophic forgetting when they are only trained on the set of points labeled at a given round. In the plot, we use task $n$ to denote the set of points that were labeled at AL round $n$.
>
> Note that each task here is actually a subset of the training set and not part of the test set. The fact that the accuracy of each task is lower than the accuracy of the test set demonstrates that each of the task distributions is distinct from the true data distribution, and the precipitous drops in accuracy for each task are evidence of catastrophic forgetting.
>
> The accuracy at round $n$ is reported after training on the newly labeled points, and the model was not trained on any dataset prior to this experiment. We believe that part of the confusion about pretraining may arise from the fact that the x-axis in figure 1 is zero-indexed, which we have since updated.
>
> We have added some updates and clarifications to this section in the revision. Please let us know if you have any further ideas on how we can improve the clarity of Figure 1 and Section 3.3.
>
> - Clarity issues: What are reasonable choices of $m^{\prime}$?
>
> What we mean by “reasonable choice of m’ ” is that the value of m’ must be commensurate with m.  Otherwise, the expected number of passes made from each sample in the history will not be smaller than it would be if we iterated through the entire dataset. The specific values of m’ are all listed in the supplementary section.
>
> - How is selection of $m’$ samples to create $B_{replay}$ done for CAL-MIR?
>
> As written in the last sentence of the section where CAL-MIR is described,  we first randomly subsample $C$ samples from $\mathcal{D}_{1:t-1}$ where $C > m’$ and compute the MIR scores only on the subset. The reviewer is correct in pointing out that the cost of computing the MIR score for all samples in history would be prohibitively expensive.
>
> - How is selection of alpha and beta done for section 4.2?
>
> All hyperparameters (including alpha and beta) in section 4.2 are identical to those used in section 4.1. These are all selected based on which values attain the highest validation set performance and the specific values are listed in the appendix.
>
> *We once again thank the reviewer for their feedback and hope that the rebuttal clarifies the questions raised by them. We are happy to discuss this if you need any further clarifications.*

---

> > ### Comment · Reviewer_HMvu · 2022-11-27
> > **Thank you for your response**
> >
> > I thank the reviewers for their response. In some of their responses and revisions the authors have answered my concerns, hence I will raise my score. However regarding my main concern, Im still not fully convinced that the degradation in performance with ER is so conclusive: most of the results dont support it, as far as I checked. It is more a matter of ones preference whether a degradation of 1% is significant versus an acceleration increase of 20%, in particular, since the authors don't provide error bars with their results.

---

> > > ### Author Response · Authors · 2022-11-27
> > > **Thank you for the response**
> > >
> > > We thank the reviewer once again for their helpful comments. The degradation in performance for ER is generally statistically significant and can be much more than 1% in some datasets (such as MedMNIST). While error bars aren't included in the main paper, the standard deviations are included in Appendix A (we will indicate this in Section 4 in the camera ready version). However, we generally agree that it mostly a matter of user preference to determine how much of a degradation in test time performance is tolerable depending on the use case. This is why we ensure that there are at least some CAL techniques that do not incur any drop in performance as well.

---

### Official Review · Reviewer_wt23 · 2022-11-04

**Confidence:** 2
**Correctness:** 1
**Technical Novelty And Significance:** 2
**Empirical Novelty And Significance:** 1
**Recommendation:** 1

**Clarity, Quality, Novelty And Reproducibility:**

The paper is well written, the motivation is very clear.

The main concern is the seemingly poor review of related work and previous approaches. There are several papers that include the same terms active, continual, learning in their titles. Even though they can have different research goals, they previously considered CL and AL in the same contexts. I do not think that one can state the contribution “We first demonstrate that active learning can be viewed as a continual learning problem and propose the CAL framework” and, at the same time, do not even refer previous papers like "Continual Active Learning for Efficient Adaptation of Machine Learning Models to Changing Image Acquisition", "Continual Active Learning Using Pseudo-Domains for Limited Labelling Resources and Changing Acquisition Characteristics" by Perkonigg et al. and "Active, Continual Fine Tuning of Convolutional Neural Networks for Reducing Annotation Efforts" by Zhou et al.

The idea of similarities between the CL and AL is not novel itself, and given that the idea of the paper is straightforward at some extent, the review of previous work should be definitely improved. At the current point, it is very difficult to asses the novelty of the paper.

**Strength And Weaknesses:**

+ The paper is well written, the motivation is very clear.
- The main problem is the insufficient review of related work and previous approaches. I need not only a citation of previous work, but also a thorough review of the novelty of the proposed algorithm over the existing literature.

**Summary Of The Paper:**

Authors state that the efficiency of traditional active learning (AL) should be significantly increased using approaches from continual learning (CL), when we consider past datapoints already used for training in AL as previous tasks. Authors test several standard CL approaches and newly proposed CL methods specifically designed for AL in the AL setting. On datasets FMNIST, CIFAR-10, MedMNIST, and Amazon Polarity, author demonstrate the ability to accelerate AL up to 2-6 times in some settings.

**Summary Of The Review:**

This paper is not ready for publication.

---

> ### Author Response · Authors · 2022-11-18
> **We thank the reviewer for their comments.**
>
> We thank the reviewer for their comments. First of all, "Continual Active Learning for Efficient Adaptation of Machine Learning Models to Changing Image Acquisition" and "Continual Active Learning Using Pseudo-Domains for Limited Labelling Resources and Changing Acquisition Characteristics" both refer to the same work by Perkonigg et al. Moreover, **we cite both Perkonnig et al. and Zhou et al. in Section 2 and discuss how CAL is distinct from either of them in the original submission**.
>
> As we state in the manuscript, the settings of the mentioned papers are very different from the CAL setting that we propose in our work despite having similar titles. Both works by Perkonnig et al. use “continual active learning” to refer to the problem of performing active learning on a data stream which is distinct from using continual learning to accelerate active learning.  Zhou et al. do not even use continual learning but instead use transfer learning to fine-tune a pre-trained model to a specific task, which is an unconventional use of the term and has little overlap with the CAL setting.
>
> We never claim that this is the first work to consider active learning and continual learning in the same setting as this would not be accurate as the reviewer points out. However, we are the first to work, to the best of our knowledge, to demonstrate that active learning can be accelerated by continual learning.
>
> *We once again thank the reviewer for their feedback and hope that the rebuttal clarifies the questions raised by them. We are happy to discuss if you need any further clarifications.*

---

### Author Response · Authors · 2022-11-19
**We have updated the manuscript with the suggested changes.**

We once again thank every reviewer for their time and feedback. We are glad that they found this problem to be well-motivated and the paper to be well-written. We have incorporated changes in our manuscript, as suggested by the reviewers.

Regards,

Authors

---

### Decision · Program_Chairs · 2023-01-20

**Decision:**

Reject

**Justification For Why Not Higher Score:**

The reviewers identified several issues which remained unresolved even after the authors' response and discussion.

**Justification For Why Not Lower Score:**

N/A

**Metareview: Summary, Strengths And Weaknesses:**

This paper presents a method to accelerate active learning. Standard active learning requires retraining the model from scratch in every round after acquiring the additional annotated data. This paper proposes using ideas from continual learning to make retraining efficient as well as less forgetful as the model observes more and more annotated data (possibly from a shifting distribution).

While the paper is indeed looking at an interesting and important problem, the reviewers raised a number of concerns related to the experiments (results as well as the setting), lack of clarity, and novelty in general (note: one of the reviewers gave a score of 1 but their main point of concern was missing literature; this review was discarded from the consideration).

The authors provided a response which was discussed. However the concerns still remained and it was felt that, in its current form, the paper does not meet the acceptance criterion. The authors are advised to take into account the reviewers's feedback to improve the paper and resubmit to another venue.